# Intransitiveness: From Games to Random Walks [†]

**Alberto Baldi [1,‡] and Franco Bagnoli [1,2,*,‡]**

[1]  Department of Physics and Astronomy and CSDC, University of Florence, via G. Sansone 1,
   50019 Sesto Fiorentino, Italy; alberto.baldi1@stud.unifi.it
[2]  Istituto Nazionale di Fisica Nucleare, sez. Firenze, Via G. Sansone 1, 50019 Sesto Fiorentino (FI), Italy
[*]  Correspondence: franco.bagnoli@unifi.it
[†]  This paper is an extended paper presented at the 6th International Conference on Internet Science, INSCI
   2019, Perpignan, France, 2–5 December 2019.
[‡]  All authors contributed equally to this work.

**Abstract:** Many games in which chance plays a role can be simulated as a random walk over a graph of possible configurations of board pieces, cards, dice or coins. The end of the game generally consists of the appearance of a predefined winning pattern; for random walks, this corresponds to an absorbing trap. The strategy of a player consist of betting on a given sequence, i.e., in placing a trap on the graph. In two-players games, the competition between strategies corresponds to the capabilities of the corresponding traps in capturing the random walks originated by the aleatory components of the game. The concept of dominance transitivity of strategies implies an advantage for the first player, who can choose the strategy that, at least statistically, wins. However, in some games, the second player is statistically advantaged, so these games are denoted "intransitive". In an intransitive game, the second player can choose a location for his/her trap which captures more random walks than that of the first one. The transitivity concept can, therefore, be extended to generic random walks and in general to Markov chains. We analyze random walks on several kinds of networks (rings, scale-free, hierarchical and city-inspired) with many variations: traps can be partially absorbing, the walkers can be biased and the initial distribution can be arbitrary. We found that the transitivity concept can be quite useful for characterizing the combined properties of a graph and that of the walkers.

**Keywords:** transitivity; random walk; Penney game; network theory

## 1. Introduction

Games are an integral part of all cultures and are one of the oldest forms of social interaction. Many games use stochastic elements (dice, coins, wheels) and most of them are based on betting for the occurrence of a given pattern, which can be just a single number (like in Roulette, Head or Tail) or more complex configurations like in card games.

It is possible to formalize some of the simplest games as a random walk on the graph of all possible patterns, so that the victory corresponds to reaching a given node.

Although some games are completely determined by the sequence of aleatory events, like in the Game of the Goose, in others the player may employ a strategy (which can simply be the choice of the target) so to increase his/her chance of success. For instance, in the game of betting on the sum of two dice (dice roulette), the strategy of betting on the seven is surely advantageous, in statistical terms.

In many such games, there exists an optimal strategy, generally for the first player. In the dice roulette, if the two players are not allowed to bet on the same exit number, the first player is surely favoured by betting on the seven.

A game is called transitive if the fact that strategy A beats strategy B and that B beats C leads for sure to A beating C. Games in which the first player is favoured are transitive.

There are also non-transitive (intransitive) game. A typical example is the game rock–scissors–paper: rock beats scissors, scissors beats paper, paper beats rock. Games in which the second player can always find a strategy (a location on the graph) which is advantageous once that the first player has chosen his/her strategy/location are non-transitive, since there can be a set of locations that are surely disadvantageous, but there cannot be one which is always advantageous, so at the end there should be a "core" of locations which are mutually advantageous, like in the rock–scissors–paper game.

In previous articles, a non-transitive game, the Penney Game [1], was investigated [2,3]. The Penney Game (as other games) can be reformulated as a random walk on a specific network, where the nodes represent the possible choices which the players can bet on and the walk is given by the sequence of random extractions. When the betting choices have been made, the selected nodes become targets and can be represented by absorbing traps for the random walker.

As usual in the study of stochastic systems, we are not interested in a single game but rather in average quantities over a statistical ensemble, so we shall speak of a set of walkers that, starting from a given probability distribution, explore all possibilities.

It is therefore possible to extend the concept of intransitiveness, and the derived observable quantities, to generic Markov chains and in particular to random walks on networks, providing a new useful tool for characterizing the networks itself, the positioning of traps and the displacement process [2,3].

The problem of random walks in the presence of traps has been deeply investigated. There are two main approaches: the study of the influence of transient traps (i.e., the walker is stopped for a certain amount of time) [4–7] and the survival probability in the presence of irreversible traps (in this case the walker disappears when meeting the trap) [8]. The problem has been extended to partially absorbing traps [9], quantum walkers [10,11], complex networks [12,13] and also to moving traps [14,15], hunting (traps pursuit walkers) with evasion [16,17], etc.

However, there are few studies about the competition among traps [2], i.e., which is the best strategy for gathering more walkers than the opponent. We suggest that this subject, which was inspired by the theory of games, can be of interest also for the community of scholars studying more general random walks and stochastic processes.

Possible direct applications concern the placement of shops in streets, where customers are approximated by random walkers and shops corresponds to traps, and the search for information by web crawlers, but in general, all processes aiming at some targets subjected to random events can profit from this approach.

The intransitiveness property is strictly related to the directionality of the displacement graph. We therefore start exploring the simplest example of directed network: a one-dimensional lattice with periodic boundary conditions.

The network topology in general plays a key role in defining the strategic positioning of the trap nodes in order to capture the most of random walkers, but the transitivity of a system is also related to absorbency of traps, i.e., the ability of players to recognize their victory. For this reason we shall also explore the effect of partially absorbing traps.

These first two models are at the same time simple and representative of the main characteristics of intransitive games. However, the devised formalism is general and we only need the adjacency matrix of the network to fully describe the competitive behaviour. Consequently, it can be applied to many different cases, from competition for the best positioning of a shop in a urban network, to the visibility of sites in the Internet. We explored this last case in the last section, where we analyze scale-free, hierarchic and city-like networks.

We shall introduce transitivity in Section 2, and the application of this concept to Markov chains in Section 3. Applications of the definitions to several problems are reported in Section 4. Conclusions are drawn the the last Section.

## 2. Transitivity

Let us start defining the concept of transitivity of a binary relation, defined on a set $\mathcal{I}$. The relation $\sim$, is transitive if

$$A \sim B, B \sim C \Rightarrow A \sim C \qquad \forall A, B, C \in \mathcal{I},$$

and a relation is intransitive if this condition is violated for at least a triplet $A, B, C$.

A relation is completely intransitive if

$$A \sim B, B \sim C \Rightarrow A \nsim C \qquad \forall A, B, C \in \mathcal{I},$$

where "$\nsim$" denotes the fact that $A$ is not in relation with $C$. Intransitive relations are not uncommon, an instance of "being father of" is completely intransitive, while "being behind" is simply intransitive.

It is possible to reformulate a game as random walk on a directed network where the nodes represent the possible choices of the players and the links describe the competitive relations.

In order to be concrete, let us recall the Penney Game [1,3], which is an extension of the heads–tails game. In the Penney Game, the two players are asked to bet not on a single coin flip, but on a contiguous succession of heads and tails, whose length $\ell$ is fixed. This gives to each player a choice among $2^\ell$ different sequences of heads and tails. The game is won by the player whose sequence will appear first. The network and the relative adjacency matrix for the case $\ell = 3$ are shown in Figure 1.

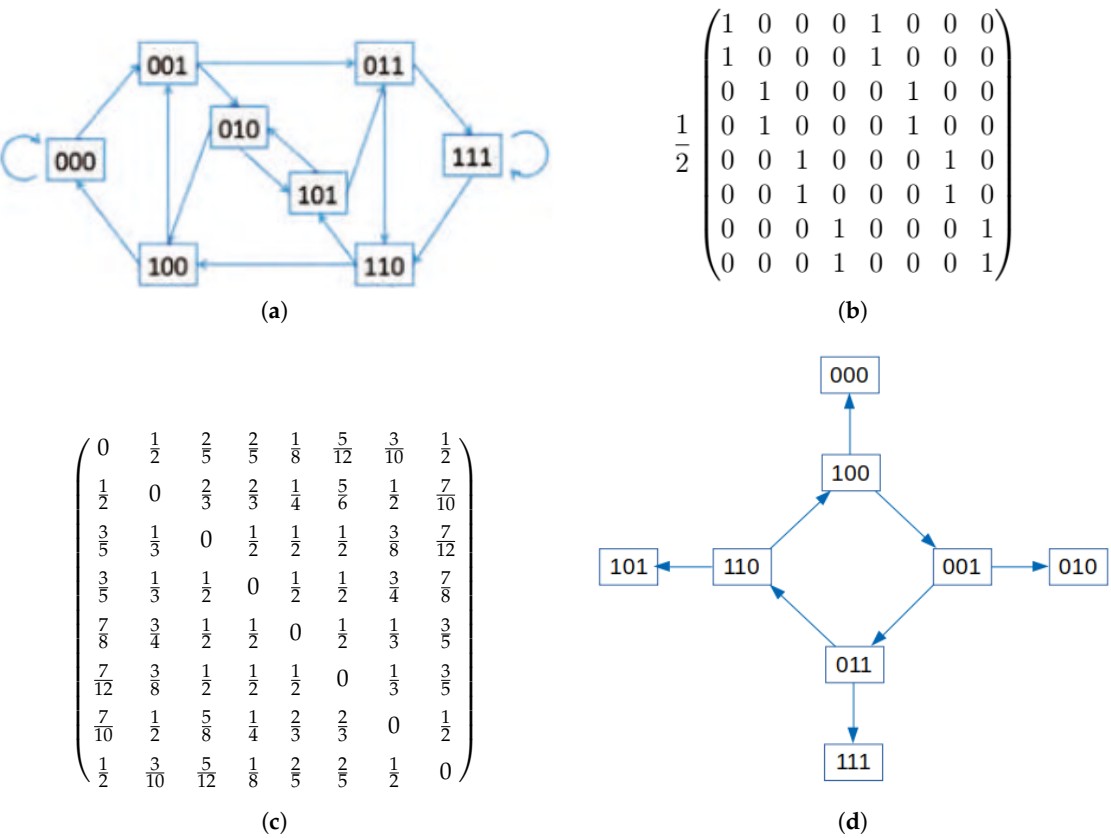

**Figure 1.** Analysis of the Penney Game with $\ell = 3$, fair coins and fully absorbing traps. (**a**) Transition graph, where every node is a possibly winning sequence, the links go from one sequence to the sequences which can be obtained with a coin toss. Each link weights $1/2$. (**b**) Weighted adjacency matrix (Markov matrix) $M$ of the system, where the sequences (indexes) are read as base-two numbers. (**c**) Victory matrix and (**d**) victory graph, where arrows mark which sequence wins with the largest probability over the given one.

By assigning a symbol to each possible sequence (for instance 1 for heads and 0 for tails, and then reading the $\ell$ sequence as a base-two number) we can map the game to a random walk on a network. Given a sequence, the next one is obtained by discarding the first digit and extracting a new one, each symbol (sequence) is connected to two other symbols (which can be the same, like in the 000 or 111 sequences). The use of a loaded coin corresponds to biased walks. The chosen sequences are absorbing traps, since, once they appear, one of the players wins.

In the Penney Game, some sequences are surely disadvantageous, but it is not evident which is the winning one. The analysis is quite simple for $\ell = 2$ and fair coins, see Figure 2. The weighted adjacency matrix (forbidding to bet on the sequence chosen by the first player) is

$$M_2(p) = \begin{pmatrix} - & 1-p & (1-p)^2 & \frac{(1+p)(1-p)^2}{1-p+p^2} \\ p & - & 1-p & 1-p^2 \\ p(2-p) & p & - & 1-p \\ \frac{(2-p)p^2}{1-p+p^2} & p^2 & p & - \end{pmatrix}.$$

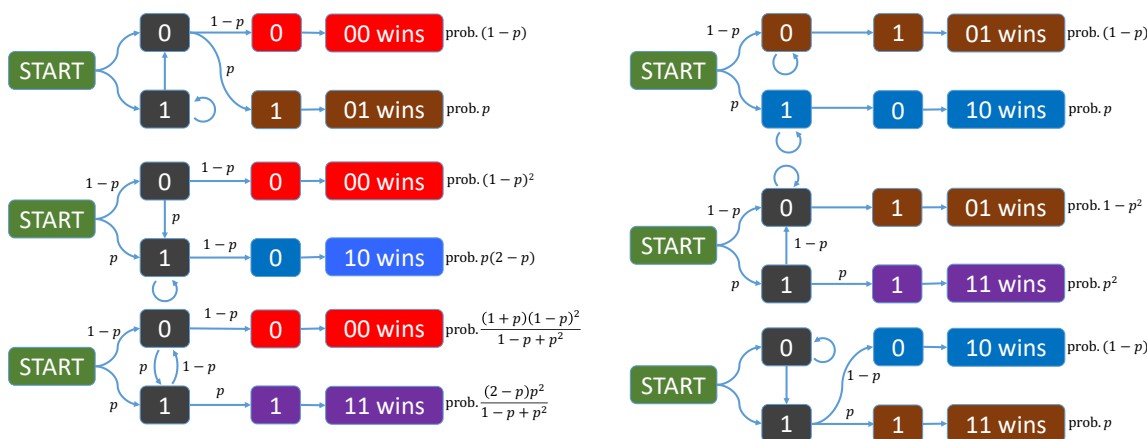

**Figure 2.** Complete analysis of the Penney Game for $\ell = 2$ and loaded coins.

The first player chooses a column (betting on the sequence corresponding to the column index in base two), and the second one chooses the row entry, corresponding to his/her probability of winning.

For fair coins, $p = 1/2$, the matrix is

$$M_2\left(\frac{1}{2}\right) = \frac{1}{4} \begin{pmatrix} - & 2 & 1 & 2 \\ 2 & - & 2 & 3 \\ 3 & 2 & - & 2 \\ 2 & 1 & 2 & - \end{pmatrix},$$

and the first player avoids columns where there is a high probability of victory for the second player, so he/she will choose sequences 01 or 10 and the second player the opposite sequence. The game is fair.

For extremely loaded coins, for instance $p = 0$, we have

$$M_2(0) = \frac{1}{2} \begin{pmatrix} - & 2 & 2 & 2 \\ 0 & - & 2 & 2 \\ 0 & 0 & - & 2 \\ 0 & 0 & 0 & - \end{pmatrix},$$

so the first player has simply to choose the sequence 00 (which will always be the only sequence appearing) to win. This holds also for finite (small) values of $p$.

However, taking for example $p = 0.51$, the matrix becomes

$$M_2(0.51) = \begin{pmatrix} - & 0.4900 & 0.2401 & 0.4833 \\ 0.5100 & - & 0.4900 & 0.7399 \\ 0.7599 & 0.5100 & - & 0.4900 \\ 0.5167 & 0.2601 & 0.5100 & - \end{pmatrix},$$

so the first player cannot choose a column for which there is no entry greater than $1/2$, and in this case it is the second player that statistically wins.

For $\ell \geq 3$ and fair coins (and also for $\ell = 2$ and slightly loaded coins) the situation is different: given any choice for the first player, the second one can always find a sequence that statistically appears before the chosen one [1].

In order to quantify the degree of intransitiveness, we introduce the victory matrix $V_{ij}$, which gives the probability that choice $i$ (second player) will win against $j$ (first player).

So, for each value of $p$ and slightly unfair coins ($p = 0.51$), one can see that if the first player chooses 00, the best for the second player is to choose 10, which also beats 01, while against 10 it is better to choose 11, and against 11 the sequence 01. This defines a victory graph, as reported in Figure 3. This graph exhibits a central loop, which is a mark of the intransitiveness of the game.

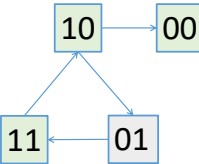

**Figure 3.** Victory graph for the Penney Game with $\ell = 2$ and loaded coins ($p = 0.51$).

A similar analysis is reported in Figure 1 for the fair Penney Game with $\ell = 3$. The corresponding victory graph (which sequence is most advantageous given a chosen one) is also represented in Figure 1. This graph shows a central loop, signalling that for $\ell = 3$ the fair Penney Game is intransitive.

We define the intransitiveness index $\sigma$ as

$$\sigma = \min_{i=1\dots N} \left\{ \max_{j=1\dots N} V_{ij} \right\} - \frac{1}{2}, \tag{1}$$

The quantity $\max_{j=1\dots N} V_{ij}$ for a specific $i$ corresponds to the maximum winning probability given the $i$ move. Then, taking the minimum over index $i$ we find the worst case, i.e., the least winning probability for the player. If this minimum is larger than one half, given the choice of the first player, there is always a sequence that statistically wins. Therefore, one cannot have a sequence that beats all the others, corresponding to the fact that the victory graph shows a loop.

Summarizing, if $\sigma > 0$, the system is intransitive and the second player statistically wins; if $\sigma = 0$, the two players can "tie", because the system permits at least two equivalent strategies; if $\sigma < 0$ the first agent has the possibility of making the optimal choice, and he/she statistically wins against the second.

One can also suppose that the winning sequence is not always recognized, corresponding to the case of partially absorbing traps.

## 3. Intransitiveness in Markov Chains

The previous analysis holds for any Markov chain, since we can associate a random walk to any Markov matrix $M$, jumping from an index $j$ to $i$ with a probability $M_{ij}$. The chain $M$ may be also derived by a real random walk on a network, identified by its adjacency matrix $A_{ij}$. In the case in

which at each node all output links may be followed with equal probability (corresponding to the fair coin in the Penney Game), we have

$$M_{ij} = \frac{A_{ij}}{k_j},$$

where $k_j = \sum_i A_{ij}$ is the output connectivity of node $j$. However, the study can be extended to biased walks (loaded coin). We also want to consider the case of partially absorbing traps.

Let us denote by $P_i(t)$ the probability of finding a walker on node $i$ at time $t$. The time evolution of the probability distribution $P(t)$ is given by

$$P(t+1) = MP(t) \qquad \Longleftrightarrow \qquad P_i(t+1) = \sum_j M_{ij}P_j(t). \tag{2}$$

In order to insert the presence of the traps, we use two distributions, $P(t)$ which gives, as before, the probability of finding the walker on the network, and $Q(t)$ for keeping track of the probability that the walker has fallen into a trap. We also need a matrix $T^{[n,m]}$, which is zero except for the indices marking the location of the traps, here in position $n$ and $m$: $(T^{[n,m]})_{nn} = (T^{[n,m]})_{mm} = 1$.

The time evolution of $P$ and $Q$ is given by

$$P(t+1) = M(\mathbb{I} - \varepsilon T^{[i,j]})P(t),$$
$$Q(t+1) = Q(t) + \varepsilon T^{[i,j]}P(t),$$

where $\varepsilon$ gives the absorptivity of traps.

If $M$ is non-singular and at least one of traps is located in the recurrent part of the chain, in the infinite-time limit all walkers fall into one or the other trap, so, denoting $\check{P} = \lim_{t\to\infty} P(t)$ and similarly for $Q$, we have $\check{P} = 0$, $\sum_i \tilde{Q}_i = 1$ and the victory matrix element $V_{ij}$, which is the probability of walkers falling in the trap at node $i$ without falling in the trap at node $j$, is simply $\tilde{Q}_i$. By repeating the procedure for all pairs of indices, one computes $V_{ij}$ and $\sigma$. If both traps are located in the transient part of the chat, $\check{P}$ is not null. In this case, by definition, both entries of the victory matrix are zero.

The victory matrix also depends on the initial distribution $P(0)$. While for games like the Penney Game, it is natural to start from a uniform distribution (at beginning no coin has been extracted), there are variations (like the Texas Hold'em poker) in which one has a previous information. For walkers in cities and web crawlers, it may happen that the starting point is predefined.

## 4. Applications

The victory graph of the Penney Game (Figure 1) shows a cycle at the core of its structure. In Ref. [3], we analyzed the transitive properties of random walkers on cycles, considering the influence of biased choice, partially absorbing traps, and the influence of the initial distribution. Simulations show that, for uniform initial distributions (all starting locations are possible), fully absorbing traps and unbiased choices (left-right displacements equally probable), $\sigma = 0$, but as soon as the process is biased it becomes intransitive, i.e., it is possible to choose the position of a trap that "obscures" the other.

If the initial distribution is more localized (uncertainty is reduced), the transitivity region of the system widens, i.e., one needs higher biases to recover the possibility of obscuring the other trap. This result is quite obvious: if one knows the starting position of the walker and that they tend to go in one direction, the first player can place his/her trap in the appropriate neighbouring position and win. On the contrary, if all positions are equivalent and there is a drift, once that the first player has chosen a position, the opponent can place his/her trap upstream.

The role of partial absorbance of traps is also that of weakening the intransitiveness, in a circle: if a walker can pass several times over a trap, the shading effect is lowered and the parameter $\sigma$ diminishes towards zero.

However, in more complex networks, more interesting phenomena may appear. While for cycles, increasing the value of the absorbancy of traps ($\epsilon$) always produces an increasing of $\sigma$ (that is always positive), for networks, there may be an intermediate value of $\epsilon$ for which $\sigma > 0$. This effect was shown for instance in Ref. [3], studying a small scale-free network (100 nodes). In this case, by varying the absorbancy $\epsilon$ of traps, one observes an intermediate intransitive regions.

In order to clarify the origin of this effect, which is due to the influence of "leaves" on the core cycle, let us consider a small network with two loops, see Figure 4b,c), and a uniform initial distribution. For very high absorbancy, it is clear that node 1, which has a larger "basin", dominates over all other position of traps. On the contrary, for very small absorbancy, it is node 2 that dominates, since all walkers from node 1 go first to node 2 and the to node 3, from where half go back to node 2 and half to node 2. For exactly vanishing absorbancy $\epsilon$, the invariant distribution is given by $x_2 = x_3 = 2/5$ and $x_1 = 1/5$, so for small but not null $\epsilon$, nodes 2 and 3 dominates over node 1, which however still has the larger basin, so most of walkers pass through node 1 and end in the "core" of nodes 1, 2, 3, passing first from node 2, which, therefore, has an advantage over node 3.

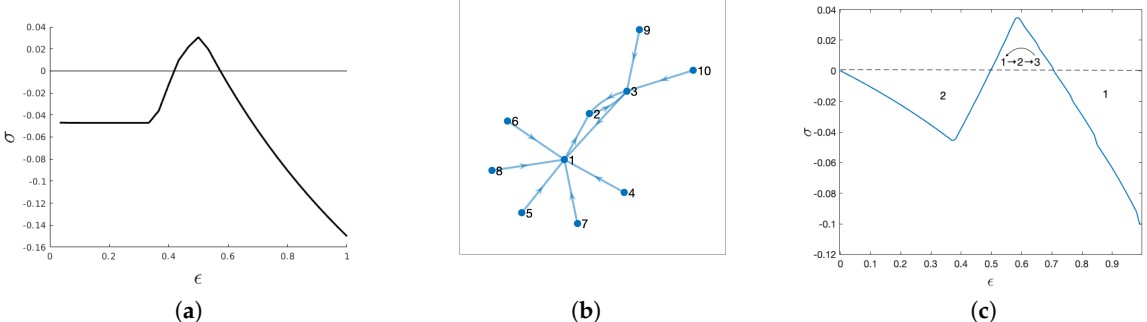

(**a**)　　　　　　　　　　　　　　　　(**b**)　　　　　　　　　　　　　　　　(**c**)

**Figure 4.** (**a**) Intransitiveness index $\sigma$ vs absorbency $\epsilon$ for a scale-free network with 100 nodes as a function of the absorbency parameter $\varepsilon$. There is intransitive behaviour for intermediate values of this parameter. (**b**) The small subnet presumably responsible for the the intransitive region $\sigma > 0$ (**c**) Intransitiveness index $\sigma$ vs absorbency $\epsilon$ for this subnet.

Increasing $\epsilon$, we have a transition of dominance from node 2 to node 1, passing through a region in which the system is intransitive, with nodes 1, 2 and 3 that may capture most of walkers, depending on which is the second player.

The next subject concerns hierarchical matrices, which are good models for communities. The network is defined by blocks, as indicated in Figure 5a [18]. In each element of a block $k$, of size $l_k$, there is a one (connection) with probability $p_k$. As shown in Figure 5b, if communities are well isolated ($p_3 \simeq 0$) and for enough absorbing traps, there is a region of intransitiveness.

Since these matrices are in average symmetric, one does not expect intransitiveness. However, when the connection among communities is very low, $\sigma$ becomes positive. This is due to the fact that for such low values of $p_3$ (and absorbance high enough) there are very few connection among communities, so that the effective network is almost always asymmetrical.

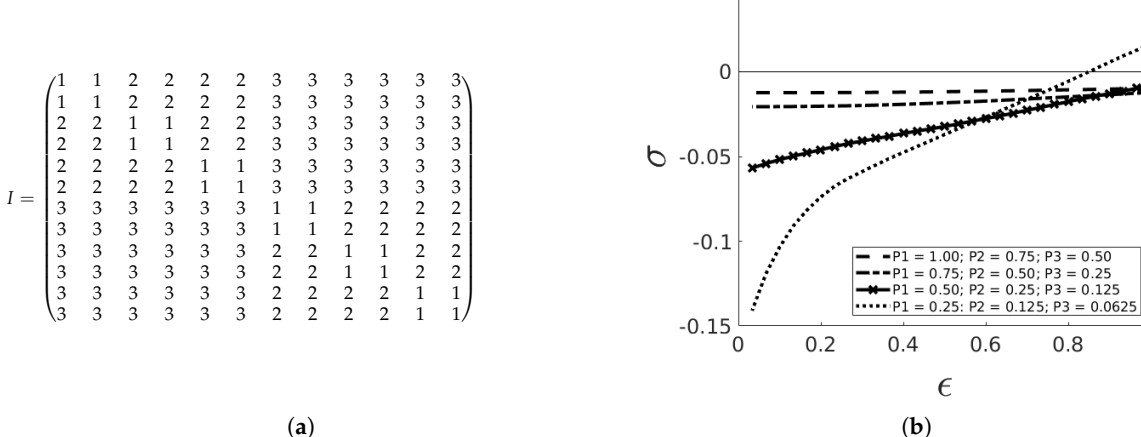

(**a**)                                    (**b**)

**Figure 5.** (**a**) Example index matrix for hierarchical matrices (communities $l_1 = 2, l_2 = 6, l_3 = 2$): for an entry $k = I_{ij}$, the corresponding adjacency matrix has ones with probability $p_k$. (**b**) Average intransitiveness index $\sigma$ vs absorbency $\epsilon$ for hierarchical communities ($l_1 = l_2 = l_3 = 3$, average over 1000 simulations).

Finally, we examined the case of a "square" city, Figure 6a, with streets that have a probability $p$ of being two-ways, and $1 - p$ of being one-way, in alternation. For small enough values of $p$, and large absorbency we have intransitiveness, Figure 6b, again stating the importance of cycles.

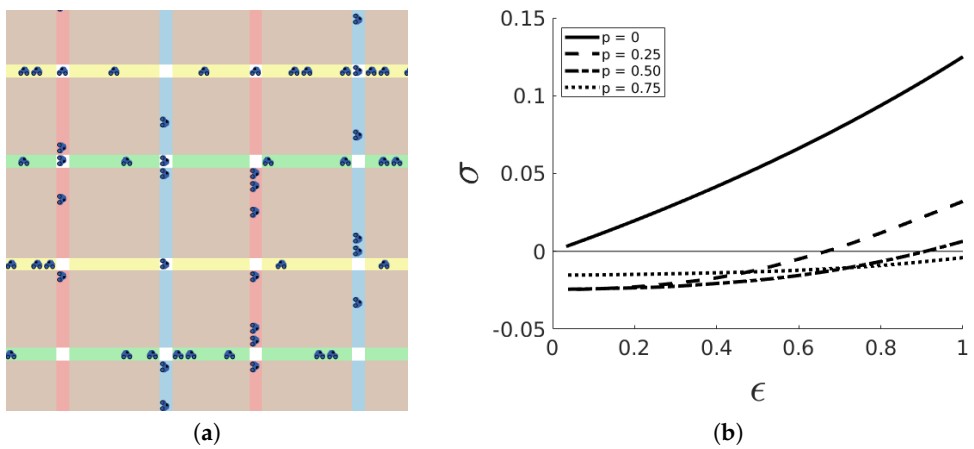

(**a**)                                    (**b**)

**Figure 6.** (**a**) A "square" city, where streets are alternating one-way ($p = 0$). (**b**) Average intransitiveness index $\sigma$ vs absorbency $\epsilon$ for a "square" city, where $p$ denotes the probability of having two-way streets instead of alternating one-way (city with $4 \times 4$ streets, average over 1000 simulations; the results are largely independent on the city size).

An interesting phenomenon occurs with reflecting boundary conditions, instead of periodic ones, and traps with different absorbancy, as reported in Figure 7. In this case one observes an intransitive region for intermediate values of the absorbency of one trap. Further investigations are needed to understand the reason for this behaviour.

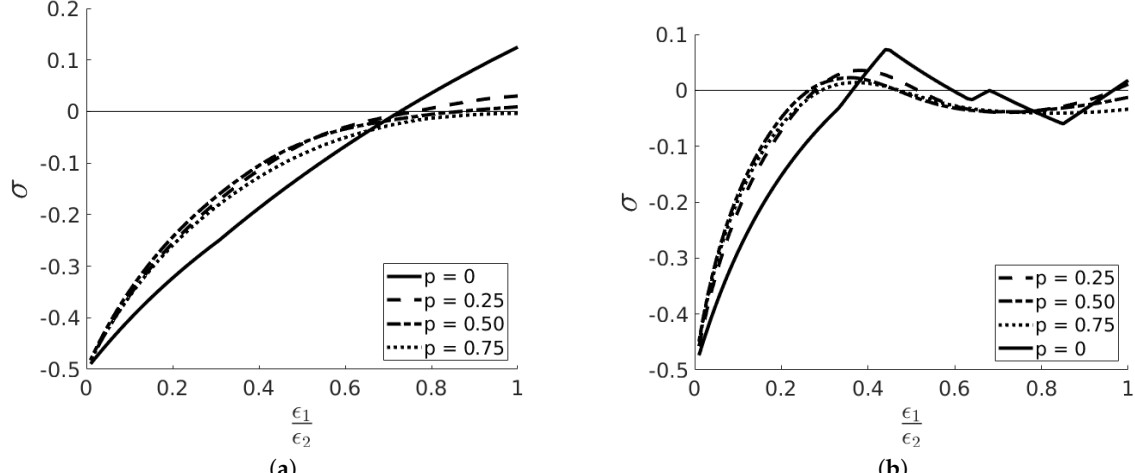

**Figure 7.** (**a**) "Square" city with periodic boundary conditions and different absorbancy ($\epsilon_1 = 1$). For large enough asymmetry $p$, there is an intransitive region near ($\epsilon_2 = 1$). (**b**) Same city but with reflecting boundary conditions, here one observes an intransitive region for intermediate values of $\epsilon_2$. The curve with $p = 0$ is different from other since in this case one doe not have to average over different realizations (1000) of the choice of one-way streets.

## 5. Conclusions

We have investigated the transitivity concept, first developed for games and generalized to Markov chains, for random walks on several networks: cycles, scale-free, hierarchical and city-inspired.

We found that the intransitiveness index can be quite useful for characterizing the properties of a graph, the influence of the initial distribution of walkers, and the different absorbancy of traps, showing that these parameters may trigger the appearance of intransitive regions in non-trivial ways.

**Author Contributions:** Conceptualization, A.B. and F.B.; formal analysis, F.B.; investigation, A.B.; methodology, A.B. and F.B.; project administration, F.B.; software, A.B. and F.B.; supervision, F.B.; writing—original draft, F.B.; writing—review & editing, A.B. and F.B. All authors have read and agreed to the published version of the manuscript.

**Funding:** This research received no external funding.

**Conflicts of Interest:** The authors declare no conflict of interest.

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
