# Peer review of "Intransitiveness: From Games to Random Walks"

_futureinternet, doi:10.3390/fi12090151_

Round 1

Reviewer 1 Report

Dear Future Internet Editors,

I have read the manuscript "Intransitiveness: from Games to Random Walks".

In this paper the authors study the mathematical concept of transitivity in game theory. In particular, the authors consider random walk models on several networks: cycles, scale-free, hierarchical and city-inspired.

The authors use an intransitiveness index in order to quantify their results and study its functional dependence on various factors such as the different absorbency of the traps and the initial distribution of the walkers.

I believe that the results presented in this manuscript would be valuable 

for researchers in the fields of game theory and random walk models. 

I can therefore recommend publication of the manuscript in Future Internet.

However, before publication the authors should make the following important change in the manuscript:

For the benefit of the readers, the authors should provide a short explanation for the claim made in page 3:

"Of these, the sequences 01 and 10 statistically win over 00 and 11..."

Yours sincerely.

Author Response

We have added a large part with the complete analysis of the Penney game for L=2. 

Reviewer 2 Report

The Authors analyse random walks on different networks considering different traps. The paper is interesting for reading and could be of interest for the readership of the journal.

However, for non-specialist in the field it is difficult to follow since only four references are cited within the text, Refs. [1, 2, 8, 9], and one cannot conclude what is new in the current manuscript and what has been done before.

Since I am not familiar with the theory of games but very much involved in random walk theory it could be nice if the authors add appropriate references in the Introduction for the reader who is not so much involved in the theory of games, and to be clear which references to consult for further reading.

For non-specialist in the field there should be more explanation of the results and what has been done before, with appropriate citations, in order this manuscript to be of interest for wider readership of the journal.

It could be nice if the authors could provide some more information and explanations regarding the traps in the games.Are these traps have the same role as the traps in the continuous time random walk theory for diffusion?

For example, in the diffusion theory if the waiting time probability distribution function of the particle in the traps is of Poisson form then the random walk process corresponds to Brownian motion for standard (normal) diffusion. If the waiting time PDF is, for example, scale-free (of power-law form) then one observes anomalous diffusive process, which is a non-Markovian process, etc.

Therefore, the corresponding stochastic process depends on the waiting time PDF. Are there some researches in this direction in the theory of games?

It could be nice to add some comment since it could attract attention to the community working on anomalous dynamics, strange kinetics and systems with memory (there are many excellent review papers and books on continuous time random walks, anomalous diffusion, fractional kinetics,...).

Author Response

We have added a paragraph with many citations illustrating the relationship between our approach and those in literature. In general, the problem of walkers in interaction with traps was studied in the presence of transient traps (that stop the walker for a while), for instance for modelling diffusion in a random media, or survival of walkers in the presence of absorbing (or partially absorbing) traps, but for what we know, very few studies (essentially ours) have dealt with the problem of competition among traps, which can be of interest to many researchers in the field of stochastic processes. 

Round 2

Reviewer 2 Report

The manuscript is suitable for publication in Future Internet.